# Dynamic Clustering Strategies Boosting Deep Learning in Olive Leaf Disease Diagnosis

**Ali Hakem Alsaeedi** [1,*], **Ali Mohsin Al-juboori** [1], **Haider Hameed R. Al-Mahmood** [2], **Suha Mohammed Hadi** [3], **Husam Jasim Mohammed** [4], **Mohammad R. Aziz** [1], **Mayas Aljibawi** [5] **and Riyadh Rahef Nuiaa** [6]

[1] College of Computer Science and Information Technology, Al-Qadisiyah University, Diwaniyah 58009, Iraq; ali.mohsin@qu.edu.iq (A.M.A.-j.); mohammad.Aziz@qu.edu.iq (M.R.A.)

[2] Department of Computer Science, College of Science, University of Mustansiriyah, Baghdad 10069, Iraq; haideritsec@uomustansiriyah.edu.iq

[3] Informatics Institute for Postgraduate Studies, Iraqi Commission for Computer and Informatics, Bagdad 10052, Iraq; dr.suhahadi@gmail.com

[4] Department of Business Administration, College of Administration and Financial Sciences, Imam Ja'afar Al-Sadiq University, Baghdad 10001, Iraq; dr.hussam@sadiq.edu.iq

[5] Computer Techniques Engineering Department, College of Engineering and Technologies, Al-Mustaqbal University, Babil 51002, Iraq; mayas.mohammed@uomus.edu.iq

[6] College of Education for Pure Sciences, Wasit University, Wasit 52001, Iraq; riyadh@uowasit.edu.iq

* Correspondence: ali.alsaeedi@qu.edu.iq; Tel.: +964-77222-35152

**Abstract:** Artificial intelligence has many applications in various industries, including agriculture. It can help overcome challenges by providing efficient solutions, especially in the early stages of development. When working with tree leaves to identify the type of disease, diseases often show up through changes in leaf color. Therefore, it is crucial to improve the color brightness before using them in intelligent agricultural systems. Color improvement should achieve a balance where no new colors appear, as this could interfere with accurate identification and diagnosis of the disease. This is considered one of the challenges in this field. This work proposes an effective model for olive disease diagnosis, consisting of five modules: image enhancement, feature extraction, clustering, and deep neural network. In image enhancement, noise reduction, balanced colors, and CLAHE are applied to LAB color space channels to improve image quality and visual stimulus. In feature extraction, raw images of olive leaves are processed through triple convolutional layers, max pooling operations, and flattening in the CNN convolutional phase. The classification process starts by dividing the data into clusters based on density, followed by the use of a deep neural network. The proposed model was tested on over 3200 olive leaf images and compared with two deep learning algorithms (VGG16 and Alexnet). The results of accuracy and loss rate show that the proposed model achieves (98%, 0.193), while VGG16 and Alexnet reach (96%, 0.432) and (95%, 1.74), respectively. The proposed model demonstrates a robust and effective approach for olive disease diagnosis that combines image enhancement techniques and deep learning-based classification to achieve accurate and reliable results.

**Keywords:** olive disease diagnosis; image enhancement; dynamic clustering; deep learning; agricultural applications

## 1. Introduction

Olive trees are essential in many world regions, providing valuable products such as olive oil and table olives. However, olive trees are susceptible to several diseases that can significantly affect their health and productivity [1]. Early and accurate diagnosis of these diseases is critical for effective management and control [2]. Artificial intelligence (AI) finds application in a wide range of domains, playing a pivotal role in addressing various challenges across industries such as [3], agricultural [4], medical [5], and more during

their early developmental stages. Using AI with outdated or incomplete data can limit the accuracy and relevance of models. This effect hinders the smart applications' ability to make informed decisions or predictions. Enhanced images are necessary before use if the application works on a computer vision area. Image enhancement based on balanced colors and brightness finds applications in various fields, such as photography, medical imaging, satellite imaging, surveillance, and computer vision tasks [6]. In computer vision tasks such as object detection or recognition, enhancing images beforehand can improve the performance of the algorithms by making the visual features more distinguishable [7].

In the proposed model, AI is harnessed to integrate clustering and deep learning, facilitating the identification of olive diseases through a comprehensive analysis of olive leaf images. Figure 1 shows the implementation scenario that was used in the proposed model.

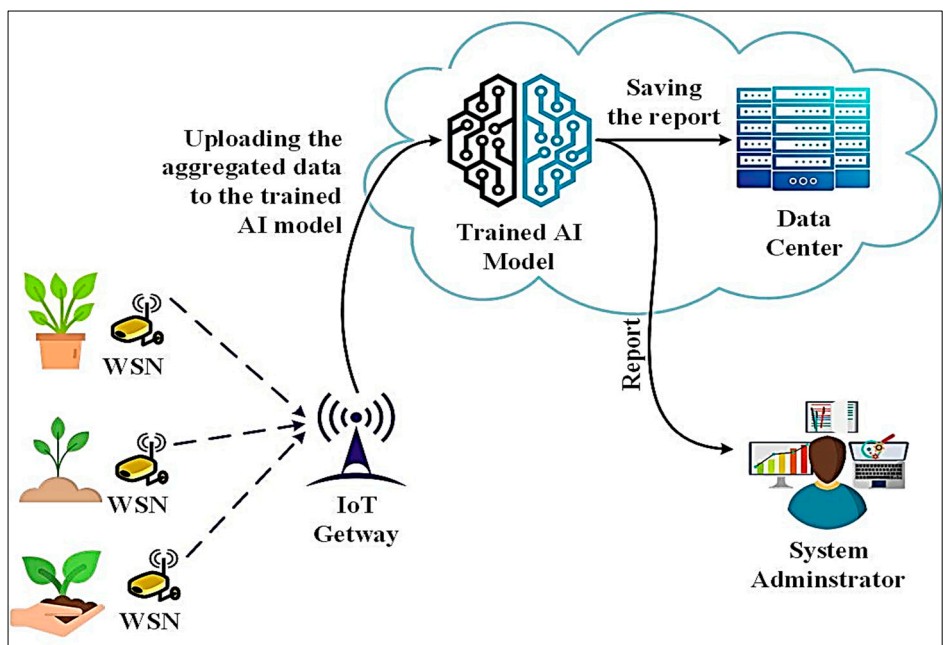

**Figure 1.** Implementation scenario of proposed diagnosis of olive disease.

The proposed scenario uses IoT technology or computer vision tools within an olive farm to collect images from sensors and imagery. The goal is to improve disease diagnosis and general monitoring of olive trees. By integrating intelligent technologies, farmers can proactively manage crop health and promptly address potential issues, contributing to more sustainable and productive farming practices. The proposed model demonstrates a robust and practical approach to olive disease diagnosis, combining image enhancement techniques and deep learning-based classification to achieve accurate and reliable results.

*1.1. Motivations*

The motivations for this model lie in addressing the challenges faced by the olive industry, leveraging AI and computer vision techniques to enhance disease detection, and providing a comprehensive and effective solution for olive disease diagnosis. Olive trees are essential in many regions, providing valuable products such as olive oil and table olives. Ensuring the health and productivity of olive trees is critical to sustaining these industries and the livelihoods of the people who depend on them. Olive trees are susceptible to several diseases that can significantly impact their health and productivity. Olive groves are exposed to various pathogens, including Aculus olearius, which can make developing an effective disease detection algorithm as challenging as creating a universal classification model [4,8]. Early and accurate diagnosis of these diseases is critical for effective management and control to prevent widespread damage. The current deep learning models used in computer vision have several drawbacks:

- Safer from unbalanced data: It can significantly affect the performance of deep learning models, leading to biased predictions and lower accuracy. A lack of sampling of minority classes can cause the model to prioritize the majority class and neglect essential patterns in the data [9,10]. Techniques such as resampling, synthetic data generation, and special loss functions are commonly used to mitigate the adverse effects of data imbalance in deep learning.
- Low accuracy with noise data: Noisy data or low-visibility images can significantly impact the accuracy of deep learning models [6]. Noise can distort underlying data patterns, leading to misclassifications and reduced performance. To address this issue, robust preprocessing techniques, noise reduction methods, and regularization strategies may enhance a CNN's ability to extract meaningful features from noisy input.
- High complexity: The size of samples in deep learning contributes to increased complexity, impacting training times and risking overfitting. Techniques such as dimensionality reduction help mitigate this challenge by improving efficiency and generalization.

*1.2. Contributions*

In this paper, the proposed model comprises several interconnected components that play critical roles in the diagnostic process. First, olive leaf images are preprocessed to remove noise and artefacts, ensuring high-quality input data. This step includes color correction. Next, the images are fed into a feature extraction component, where they are normalized and split. A deep learning-based classification network, trained on a large dataset of labelled olive disease samples, accurately identifies the disease type, such as anthracnose or leaf spot disease, based on the augmented image features. Finally, the diagnostic results are presented to the user, providing valuable insights into the health status of the olive leaves. The study aims to propose an effective model for accurate olive disease diagnosis. It combines image enhancement techniques and deep learning-based classification. The proposed model achieves several contributions, which could be summarized as follows.

- Integration of clustering and deep learning: The proposed model integrates clustering and deep learning techniques to analyze olive leaf images comprehensively. This combination of methods helps in the accurate identification of olive diseases.
- Robust diagnostic approach: The proposed model provides a powerful and practical method for diagnosis of olive disease. It aims to provide accurate and reliable results by combining image enhancement and deep learning-based classification.
- High accurate and reliable results: The employment of image enhancement through color correction methodology, coupled with intensity assessment using the hierarchical convolution approach, and applying the dynamic clustering technique, culminates in a notably precise outcome characterized by significant accuracy.
- Low complexity: Training deep neural networks on selected samples enhances the accuracy of diagnostic procedures compared to training on the entire dataset, owing to reduced computational complexities by utilizing a smaller subset of data during training and reducing the overfitting.

The paper structure is organized as follows: Section 2 deals with related work in computer vision for diagnosis, verification, and identification. Section 3 is in agreement with material and methods, while Section 4 discuses the results. Section 5 outlines the proposed model and its components.

## 2. Related Works

This section explores various studies that proposed different deep learning models for diagnosis and image classification. The development of machine learning models saw numerous studies proposing different approaches for classification and regression tasks. However, many of these research endeavors primarily relied on standard machine learning or deep learning classification algorithms, often without adequately addressing the limitations associated with these models. It is worth noting that machine learning

algorithms tend to face challenges that could be addressed with better-prepared data, higher accuracy rates, and reduced complexity in terms of space and time.

Alshammari et al. [4] proposed a deep learning model combining vision transformer and convolutional neural network architectures for accurate olive disease detection. The study combines the performance of two different deep learning architectures, namely vision transformer (ViT) and a convolutional neural network (CNN), to improve the accuracy and efficiency of disease detection in olive plants. The weaknesses of the proposed system lie in improper training, as the model was trained on imbalanced data, which may affect its learning ability and exacerbate other problems related to uneven class distribution or accurate class prediction for classes with small counts.

In [11], the study focuses on using the capabilities of convolutional neural networks (CNNs) to identify and classify different diseases affecting olive leaves accurately. By applying deep CNNs to the problem of olive leaf disease classification, the work is consistent with the trend of using advanced machine-learning techniques to improve disease detection and monitoring. Without sufficient emphasis on regularization techniques, hyperparameter tuning, or cross-validation, the proposed model may be susceptible to overfitting the training data. Overfitting could lead to poor performance on unseen data and undermine the practical utility of the model.

Ksibi et al. [12] proposed a hybrid deep learning model for olive leaf disease detection and classification. The proposed model is composed of neural networks from the ResNet50 and Mo-bileNet models. The model combines elements of the MobileNet and ResNet architectures to improve its ability to identify and categorize different diseases affecting olive leaves accurately. This approach aims to leverage the strengths of the different architectures while mitigating their weaknesses. However, its computational cost and memory requirements may limit its applicability in resource-constrained environments.

In [13], the author used the deep learning architecture of Inception V3 to accurately and efficiently classify olive leaf diseases. This framework addresses the critical challenge of identifying diseases affecting olive leaves by leveraging the capabilities of a robust convolutional network architecture. By leveraging the strengths of Inception V3, the framework aims to achieve higher accuracy in distinguishing between different types of olive leaf diseases. Despite the complexity of the framework, the model can achieve a high level of accuracy.

Gulzar [14] presents a study on using deep learning techniques for fruit classification. He used the MobileNetV2 with the deep transfer learning technique to classify fruit images. The classification layer of MobileNetV2 is replaced by a customized head, which produces the modified version of MobileNetV2 called TL-MobileNetV2. In addition, transfer learning is used to retain the pre-trained model. The study findings indicate that transfer learning significantly contributes to better results, and the use of the dropout technique helps mitigate overfitting in transfer learning.

Mamat et al. [15] proposed a deep learning model to enhance the classifying of the ripeness of oil palm fruit and recognize a variety of fruits. The authors proposed simple and effective models using a deep learning approach with You Only Look Once (YO-LO) versions. There are several issues with the proposed model that need to be considered. These include difficulties in dealing with overlapping objects, detection of partially hidden objects, and possible biases in the data used to train the model. In addition, the performance of the proposed model may be affected in certain situations due to the balance between spatial resolution and speed and its complicated structure.

## 3. Material and Methods

This section discusses the material and methods for boosting deep learning in olive disease diagnosis using dynamic clustering strategies.

### 3.1. Prerequisite

This section will explain the prerequisite techniques used in the proposed model.

### 3.1.1. Convolutional Neural Networks (CNN)

Deep learning is an essential approach in artificial intelligence as it can analyze large-scale data and predict appropriate outcomes in the real world [16]. Neural networks are a subset of machine learning centered around building artificial neural networks and training them on actual data for precise predictions [17]. These networks eliminate the need for distinct feature selection and extraction algorithms while also excelling in managing extensive datasets encompassing various forms of information, including text and multimedia files [4,18–22].

The deep learning model comprises an input, hidden, and output neuron layer. This architecture incorporates numerous intricate hidden layers to extract accurate patterns and complex features from training data. These patterns can be challenging to discern using conventional machine learning algorithms [21]. This feature enabled breakthroughs in image recognition, language processing, natural language processing, and many other fields.

CNNs are considered one of the most critical types of deep learning due to their ability to process data with a network structure (e.g., images) [17,22,23]. CNNs extract meaningful features by applying a convolutional process to learn the spatial hierarchy of the features extracted from the input data. The primary operations performed in CNNs are convolution, pooling, and fully connected layers. Figure 2 shows the simple architecture of the CNN.

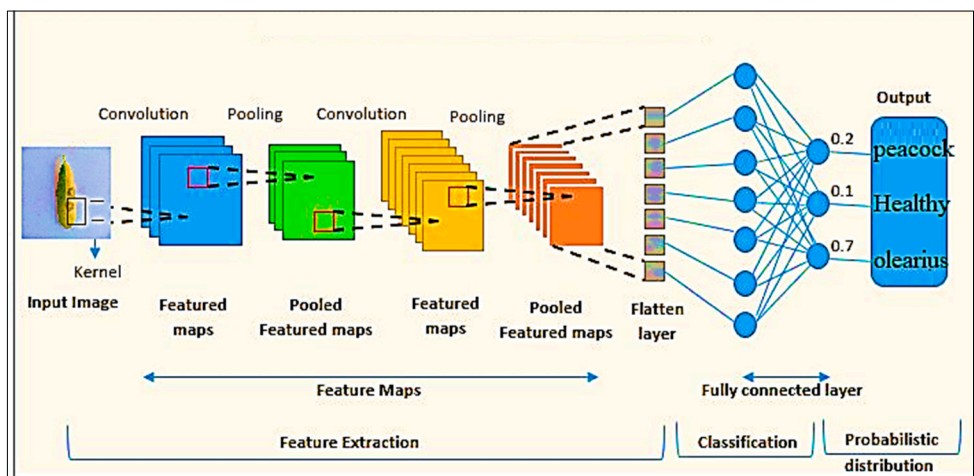

**Figure 2.** Simple CNN architecture.

The convolutional layer consists of many trainable filters. The convolution of each trainable filter over the input data produces different spatial patterns or relevant features representing the original data [24]. CNNs apply a convolution process to learn the spatial hierarchy of features extracted from the input data. The convolution process involves applying adaptive filters to the data, resulting in different patterns representing different spatial features in the input data [25].

Pooling layers are often inserted between successive convolutional layers. The key idea behind pooling layers is to down-sample the feature maps and reduce their spatial dimensions. Many techniques are used to perform the pooling layer [26]. Max pooling is a standard technique for selecting the highest value within a specific pool size. In addition, the pooling layer is beneficial for reducing computational complexity, improving translation invariance, and extracting key features.

The CNN has one or more fully linked layers. These layers allow the network to learn complicated mappings as they connect each neuron of the previous layer to the following layer. In classification problems with multiple classes, fully connected layers are often followed by an activation function (e.g., Softmax).

To achieve effective results, large amounts of data must be fed into the CNN as training data; a lack of training data can lead to overfitting, which affects the network's ability to generalize new real-world data [27]. This highlights the computational challenges of train-

ing CNNs. In addition, deploying and training CNNs requires significant computational resources, such as powerful GPUs or specialized equipment [28,29].

The architecture of a convolutional neural network (CNN) plays a critical role in its ability to process and extract meaningful features from input data effectively. The architecture of a CNN refers to the arrangement and organization of its layers and their connectivity patterns [30].

### 3.1.2. Dynamic Clustering Empowering Deep Learning

Dynamic clustering represents a powerful technique for enhancing the performance of deep learning models [31,32]. In contrast to static clustering methods, which establish fixed clusters before training, dynamic clustering adopts an adaptive approach. This approach entails making real-time adjustments to clusters during training, enabling the model to capture complex and evolving data patterns [19] effectively. The integration of dynamic clustering with deep learning offers several key advantages. Firstly, the model can selectively focus on relevant data subsets at different training stages, thereby improving feature extraction and classification accuracy, particularly for complex and diverse datasets. Additionally, dynamic clustering can help uncover new data patterns and identify outliers or novel cases, which is particularly valuable in situations where the data distribution changes over time [32].

Furthermore, the combination of dynamic clustering and deep learning solves the challenges of large datasets. By partitioning the data into evolving clusters, computational efficiency is increased, convergence is accelerated, and memory requirements during training are reduced [28]. The synergy between dynamic clustering and deep learning enables models to exploit data variability effectively, adapt to changing patterns, and achieve high-performance benchmarks in various applications, including image recognition, natural language processing, and recommender systems. As research in this area advances our understanding, combining dynamic clustering and deep learning promises to open up new possibilities in intelligent systems [33–35].

### 3.2. Dataset Description

The model under consideration is rigorously tested and evaluated utilizing a dataset available at the link "https://github.com/sinanuguz/CNN_olive_dataset (accessed 11 September 2023)" This dataset comprises a curated collection of 3400 olive leaf images meticulously gathered from the Turkish city of Denizli during spring and summer. These images are thoughtfully classified into three distinct categories: those depicting leaves afflicted by Aculus olearius, those portraying olive spot disease-inflicted leaves, and images capturing the essence of healthy leaves. The characteristics of the images in this dataset are RGB colors with dimensions of 800 × 600. Table 1 illustrates the number of samples in each class of the dataset.

**Table 1.** Summary of the dataset of olive diseases.

| Olive Disease | Train | Test | Total |
| --- | --- | --- | --- |
| Healthy | 830 | 220 | 1050 |
| aculus_olearius | 690 | 200 | 690 |
| olive_peacock_spot | 1200 | 260 | 1460 |

Figure 3 shows a representative subset of the comprehensive olive disease dataset, which includes three distinct categories: (healthy, aculus_olearius, and olive_peacock_spot). This visual snapshot provides a glimpse into the diverse range of conditions recorded in the dataset, showing healthy leaf samples and those exhibiting characteristic symptoms of olive diseases.

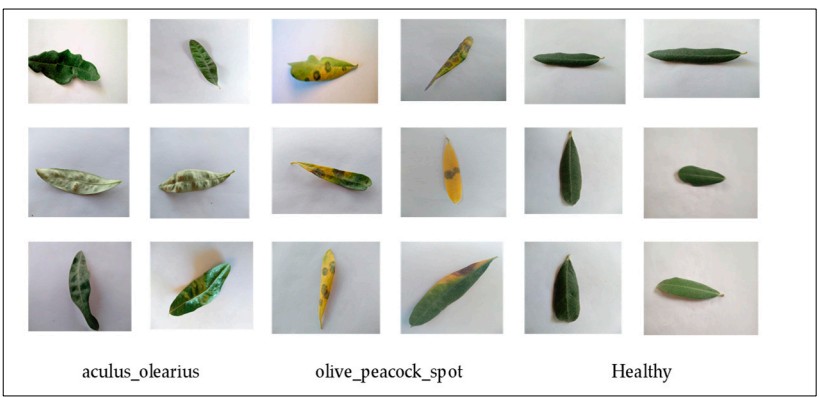

**Figure 3.** Sample of the olive diseases dataset.

*3.3. Methodology of the Research*

Figure 4 illustrates the step-by-step process of the proposed model for olive disease diagnosis. The model consists of several components, each critical in the diagnostic process. At the beginning of the pipeline, the olive leaf images are preprocessed to remove noise and artefacts and ensure input data quality. In this step, colors are corrected. After color correction, the images are fed into prepared data to extract features, normalized, and split into a deep learning-based classification network trained on a large dataset of labelled olive disease samples. Based on the augmented image features, the classification network accurately identifies the disease type, such as anthracnose or leaf spot disease. Finally, diagnostic results are presented to the user, providing valuable insights into the health status of olive leaves.

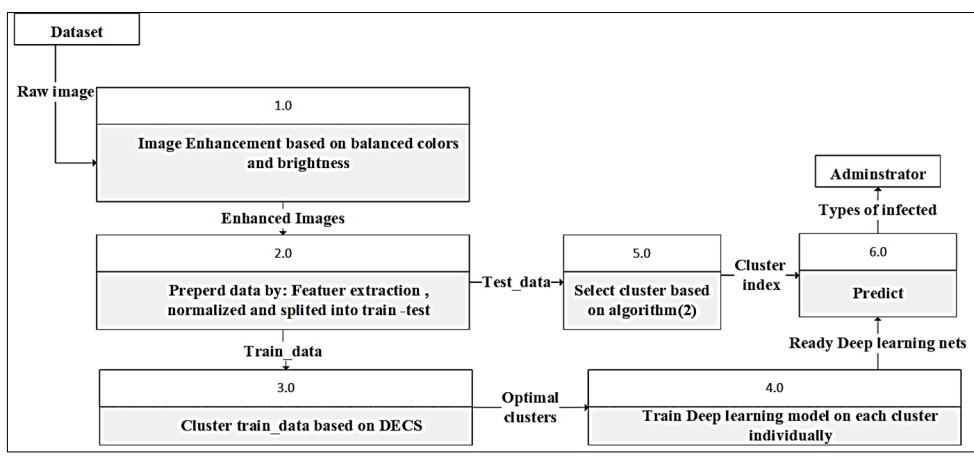

**Figure 4.** The main steps of the proposed model.

3.3.1. Image Enhancement Based on Balanced Colors and Brightness

Images may contain noise or unwanted artifacts that can affect the overall quality of the image. Noise reduction algorithms can help mitigate these problems while preserving essential details. Balanced colors and brightness are used to protect the natural look. This enhancement aims to make the image more visually appealing and easier to interpret without altering the original content. The white balance algorithm with contrast limited adaptive histogram equalization (CLAHE) enhances the input images. Before applying CLAHE, the input image color space is transferred to the LAB color space [36]. The LAB color space consists of three components: L-channel (luminance), which represents the luminance information; A-channel, which represents the color information along the green-magenta axis; and B-channel, which represents the color information along the blue–yellow axis. [37]. Therefore, the CLAHE is applied to the A and B channels separately, effectively

removing color casts and enhancing the color contrast. Algorithm 1 illustrates the processes used in the image enhancement phase.

---

**Algorithm 1**: Image Enhancement Algorithm

---

**Input**: input the BGR image (I)
**Output**: Enhanced _image

    *i.*    *I ← input the BGR image*
    *ii.*   *lab_image← BGR2LAB (I)*
    *iii.*  *l_channel,a_channel,b_channel← extract_LAB component(lab_image)*
    *iv.*  *a_channel_corrected ← CLAHE(a_channel)*
    *v.*   *b_channel_corrected ← CLAHE(b_channel)*
    *vi.*  *Enhanced _image←merge(l_channel,a_channel_corrected,b_channel_corrected)*

**Return** *Enhanced _image*

---

After using Algorithm 1, the color-corrected image displayed improved clarity and a more natural appearance, effectively bringing out the fine details of the subject while preserving the overall brightness and contrast. Figure 5 shows the comparison between the input and outcome of Algorithm 1.

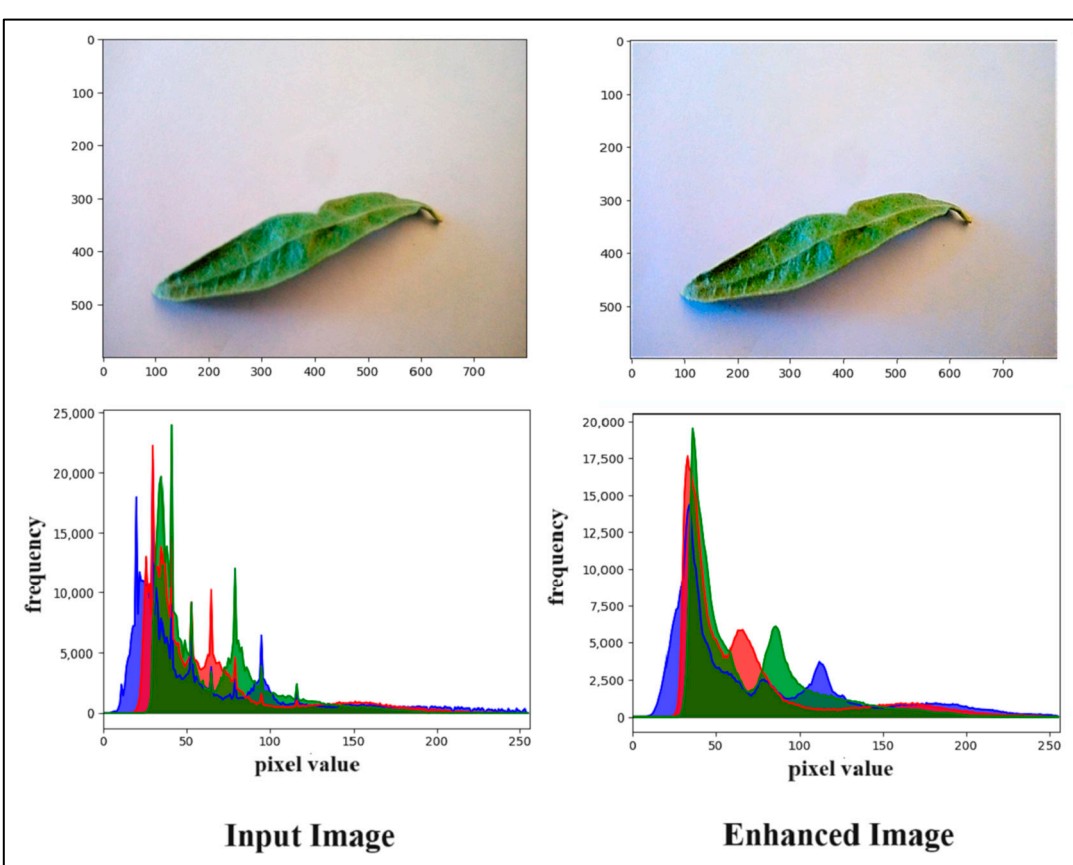

**Figure 5.** The outcome of image enhancement based on balanced colors and the brightness step.

According to the olive images, olive leaves have three types: the first type has natural leaves with green and a flat surface, while the remaining two categories have either color spots or irregularities in the leaf's surface and fade in color. Therefore, the improvement we proposed in the image increases the contrast, which can help better distinguish between these different types of leaves.

3.3.2. Prepared Data

The data preparation for olive disease diagnosis involves several essential steps, beginning with feature extraction using the convolutional phase of deep learning. In this phase, raw images of olive leaves are passed through a train convolutional neural network (CNN) to extract high-level features capturing the characteristic patterns and features unique to disease types. Let $X$ be the raw input image data, and $f()$ represent the convolutional phase of the CNN. This process encompasses a triadic sequence of convolutional layers, followed by max pooling operations, culminating in a flattening operation, collectively generating a 128-dimensional feature vector. The output of the convolutional phase, denoted as $Z$, captures the high-level features extracted from the raw image. Equation (1) represents the feature extraction process.

$$Z = f(X) \tag{1}$$

Here, $Z$ is a feature representation of the input olive leaf image, holding the learned patterns and features of the disease diagnosis. A series of experiments resulted in selecting this specific architecture for feature extraction from the input image, chosen due to its superior performance over other alternatives. This architecture uses convolution to extract features from the image. It includes multiple layers, such as convolutional layers, max pooling layers, and a global average pooling layer. The input for this architecture is an RGB image with a resolution of $800 \times 600$ pixels. It consists of three layers of convolution, each followed by max pooling. The settings for each layer are as follows:

- Input image: the first layer takes an RGB image of resolution $800 \times 600$ pixels.
- Layer 1: The convolutional layer 1 applies 32 filters to the input image using a kernel size of $3 \times 3$ and a stride of 1. The activation function used in this layer is rectified linear unit (ReLU). Max Pooling 1 takes the maximum value over a $2 \times 2$ pool size with a stride of 2—the output of this layer ($399 \times 399 \times 32$).
- Layer 2: Applies 64 filters on the input image, using a kernel size $3 \times 3$ and a stride of 1. The activation function used here is also ReLU. The second max pooling operation uses the same methodology, yielding an output of size $198 \times 198 \times 64$.
- Layer 3: The third convolutional layer employs 128 filters, using a kernel size of $3 \times 3$ and a stride of 1. Again, ReLU serves as the activation function. The third max pooling operation works similarly, providing an output of size $97 \times 97 \times 128$.
- Layer 3: This layer applies two types of max pooling. Initially, Max Pooling 4 chooses the maximum value across a $2 \times 2$ pool size with a stride of 2. The global average pooling calculates the average value for each feature map, reducing the spatial dimensions to $1 \times 1$, resulting in an output of size 128.

Figure 6 shows the network architecture used to extract the features from the input image.

After feature extraction, the extracted features are normalized to ensure data uniformity and stability. Z-score normalization, or standardization, is applied to rescale the features so that their mean is zero and their standard deviation is one. In this case, given $Z$ with dimensions m $\times$ n, where m is the number of samples (olive leaf images), and n is the number of extracted features for each instance, the normalized features norm, $Znorm$, can be calculated as in Equation (2):

$$Z_{\text{norm}} = \frac{Z - \text{mean}\,(Z)}{\text{std}\,(Z)}. \tag{2}$$

After normalization, the data were divided into training and test sets. A split ratio of 70% for training and 30% for testing ensures adequate data for model training while providing enough data for unbiased evaluation. The training set trains the CNN model to recognize patterns associated with specific diseases. In contrast, the testing set evaluates model performance and generalization to unseen data.

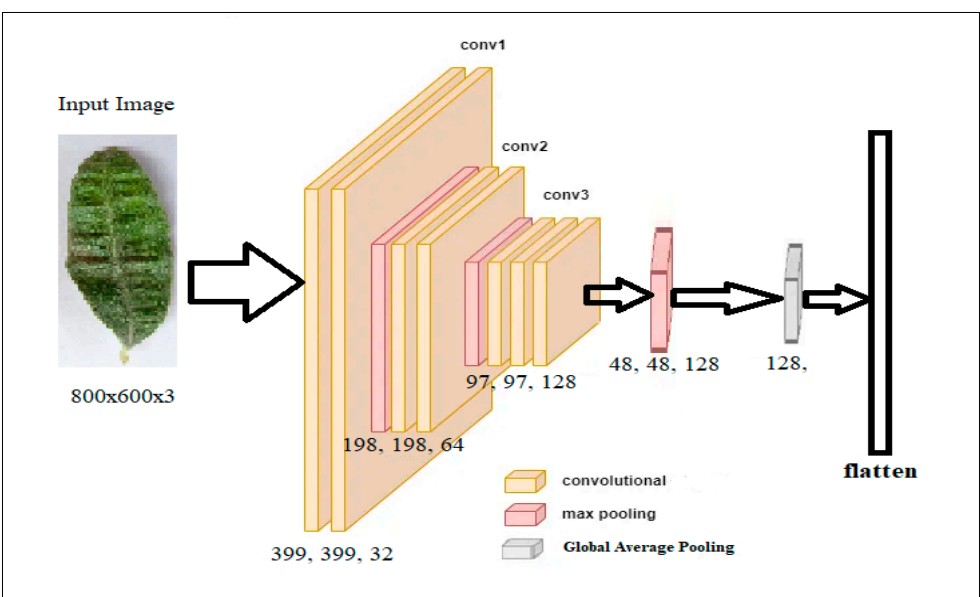

**Figure 6.** The network architecture used to extract the features.

### 3.3.3. Cluster Train_Data Based on DECS

Cluster the training data using dynamic evolving Cauchy possibilistic clustering, which operates according to the self-similarity principle (DECS). This approach groups similar olive leaf images, creating distinct clusters that effectively capture the inherent patterns and relationships within the dataset [38]. This clustering approach allows the deep learning model to better recognize and learn from the different features of the olive leaf samples, improving its ability to classify different disease types accurately. According to [38], this step has two activities: generate the initial clustering pool and optimize the existing one.

### A. Generate Initial Clustering Pool

In the beginning, the Cauchy density clustering algorithm calculates the density $\left(\gamma_i^j\right)$ of point $z(i)$ for cluster $j$, where $j$ is an index representing each cluster in the data. At each step, point $i$ is considered, and its local density value $\left(\gamma_i^j\right)$ is compared with the maximum density threshold $(\Gamma_{\max})$ [39]. If $\gamma_i^j$ is greater than or equal to $\Gamma_{\max}$, the algorithm generates a new cluster; otherwise, the point is added to a cluster that maximizes the local density $(\gamma)$. The local density of new data points is calculated by using Equation (3).

$$\gamma_i^j = \frac{1}{1 + \frac{1}{\sigma_t^2}\left(z(i) - \mu^j\right)^T\left(\Sigma^j\right)^{-1}\left(z(k) - \mu^j\right) + \frac{q(M-1)}{\sigma_t^2 M^j}} \tag{3}$$

where the center of the $j^{\text{th}}$ cluster with dimensions m × 1 is denoted by $\mu^j$. The lower and upper indices, $z_i^j$, represent the $i^{\text{th}}$ sample within the $j^{\text{th}}$ cluster. Alternatively, the cluster center notation can be expressed as $\mu_{M_j}^j$ to signify that the jth cluster comprises $M^j$ samples. Equation (4) calculates the $\mu^j$.

$$\mu^j = \frac{1}{M^j}\sum_{i=1}^{M^j} z_i^j \tag{4}$$

The covariance matrix of the jth cluster, denoted as $\Sigma^j$, has the dimensions m×m. Equation (5) is utilized to compute this covariance matrix.

$$\Sigma_{M^j}^j = \frac{1}{M^j - 1}\sum_{i=1}^{M^j}\left(z_i^j - \mu_{M^j}^j\right)\left(z_i^j - \mu_{M^j}^j\right)^T \tag{5}$$

When the density condition is met ($\gamma_1^j > \Gamma_{\text{mar}}$), the cluster undergoes an update. A new data point is added to the cluster that exhibits high cluster density. Following the addition of the new point, the algorithm (DECS) updates all cluster parameters. The cluster size (*M*) increases by one. Equation (6) is employed to compute the new center of the data.

$$\mu_{M^j+1}^j = \mu_{M^j}^j + \frac{1}{M^j+1}\left(z(k) - \mu_j\right) \tag{6}$$

The non-normalized covariance matrix (parameter *S*) states are updated using Equation (7).

$$S_{M^j+1}^j = S_{M^j}^j + \left(z(k) - \mu_j\right)\left(z(k) - \mu_{M^j+1}^j\right)^T \tag{7}$$

The covariance matrix is then determined using Equation (8).

$$\Sigma_{M^j+1}^j = \frac{1}{M^j}S_{M^j+1}^j \tag{8}$$

B. Optimize Clusters

In this phase, the DECS algorithm identifies clusters that exhibits significant correlations and forwards them to a newly created cluster pool. At the same time, the remaining clusters with lower correlations are subsequently redistributed to clusters with higher correlations. Equations (9) and (10) compute each cluster's membership, differentiating between good (high correlations) and less-favorable clusters. These equations assign a value of 1 to good clusters and 0 to the others.

$$\omega_{x_i} = \begin{cases} 1 & if \quad c_{ir} - \frac{\sum_{j=1}^n d\left(x_i, x_j\right)}{n-1} \geq 0 \\ -1 & \qquad\qquad\quad otherwise \end{cases} \tag{9}$$

In cases where $j \neq i$, $j$ represents the index of a point within the same cluster, while $c_{ir}$ denotes the nearest point to $x_i$ within a different cluster.

$$\beta_j = \begin{cases} 1 & if \quad \frac{1}{m}\sum_{i=1}^n \sum_{j=1}^m \omega_{x_j} \geq 0 \\ 0 & \quad otherwise \end{cases} \tag{10}$$

As per Equation (10), the $\beta$ values are binary (0 or 1). Clusters with nonzero $\beta$ values (1) proceed to the next clustering generation, while clusters with $\beta$ values of zero (0) necessitate reassignment to nonzero clusters. The reallocation involves computing average distances ($\delta$) between points to be redistributed and computing all nonzero $\beta$ cluster points, as denoted by Equation (11).

$$\delta = \frac{1}{m}\sum_{j=1}^m \parallel x_j - x_i \parallel^2 \tag{11}$$

3.3.4. Train the Deep Learning Model on Each Cluster Individually

The training phase involves the implementation of a deep learning model for each cluster, allowing for focused learning on specific groups of olive leaf images exhibiting standardized features. The deep learning architecture includes successive layers before the dense layers. These processes optimize information flow and reduce dimensionality, preparing the data for subsequent fully connected dense layers. This enhancement improves the model's ability to detect complex patterns and achieve accurate classifications.

3.3.5. Select a Cluster

In the cluster selection step, an innovative approach is used to determine the most appropriate cluster for a new data point. The procedure assumes that the new point serves as the hypothetical center of a cluster and calculates the distances between this new point and all other points within the cluster. The average distance is then calculated for those

points whose distance to the new center is less than the distance to the original center of the cluster.

Mathematically, let $x_n$ represent the new data point, $C_j$ denote a cluster, $c_j$ signify the center of the cluster $C_j$, and $d(x, y)$ represent the Euclidean distance between points $x$ and $y$. The procedure can be formalized as follows in Algorithm 2.

---

**Algorithm 2**: select cluster for new data point

---

**Input**: new data point x_n, set of clusters C
**Output**: Selected cluster for new data point
    *i. Calculate center distances:*
    *For each cluster $C_j$ in C:*

$$center\_distance\left(C_j\right) = d\left(x_n,\ c_j\right)$$

    *ii.Calculate average distance:*
    *For each cluster $C_j$ in C:*

$$N_j = number\ of\ points\ in\ cluster\ C_j$$
$$avg\_distance(C_j) = \frac{\sum_{i=1}^{n_j}(d(x_i,\ x_n))}{N_j}\ /\ for\ x_i\ in\ C_j\ if\ d(x_i,\ x_n)$$
$$< center\_distance\left(C_j\right))$$

    *iii.Select cluster:*
      $selected\_cluster = argmin\left(avg\_distance\left(C_j\right)\right)\ //\ for\ C_j\ in\ C$
**Return:** selected_cluster

---

### 3.3.6. Predict

After training individual convolutional neural networks (CNNs) for each cluster, the proposed model selects the index of the closest cluster. Then, the model deploys the CNN corresponding to the specified index to make predictions. This sophisticated system leverages the exceptional feature extraction capabilities of cluster-specific CNNs and improves prediction accuracy by dynamically adapting to the inherent features of the input data. The CNN architecture in the proposed model consists of four dense layers. To optimize the efficiency of the network, specific filter sizes were assigned to each dense layer to match the complexity of the task harmoniously. The first dense layer, which acts as a skilled feature extractor, has 64 units, so the model can pick up on smaller pieces of data. This layer uses $3 \times 3$ filters that promote capturing minute details while ensuring computational efficiency. Once the network begins learning, the subsequent dense layer uses 128 units and larger $5 \times 5$ filters. This larger filter size allows the model to decipher complicated patterns hidden in the data and strengthens its pattern recognition capabilities. The third dense layer consists of 256 units working harmoniously with $3 \times 3$ filters. This strategic configuration allows the model to dive deep into the nuanced interplay between different features.

Finally, the last dense layer, a central decision-making hub, consists of three units (one for each class) and comprises small but influential $1 \times 1$ filters. This dimensionality reduction not only refines the feature representation, but also simplifies the computational task. Within this complex network, activation functions determine the CNN's ability to decode complex relationships. The first layers use rectified linear units (ReLU), a robust choice that gives the model nonlinear dynamics and furthers its ability to capture complicated patterns. The last dense layer uses the Softmax activation function. This calculated move pushes the model to make probabilistic predictions for more than one class, giving it a solid basis for classification. The CNN architecture can get through complex layers, find essential features, and make accurate predictions by carefully choosing filter sizes, assigning units, and combining them with the right activation functions.

## 4. Results

This section includes a comprehensive discussion of experiment results and compares the proposed model with some recent studies on the same dataset. Accuracy, precision,

recall, F1 score, and loss function are all essential concepts in machine learning evaluation and data science.

- Accuracy is the ratio of correctly predicted observations to total observations. It measures how well a model can classify or predict data correctly. Let $TP$ be the number of true positives, $TN$ be the number of true negatives, $FP$ be the number of false positives, and $FN$ be the number of false negatives. Equation (11) calculates the accuracy.

$$\text{Accuracy} = \frac{TP + TN}{TP + TN + FP + FN} \tag{12}$$

- Precision is the ratio of $TP$ results divided by the number of all positive outcomes, including those incorrectly identified. It is especially beneficial when the consequences of $FP$ are significant, as seen in medical diagnoses where a false positive could result in unnecessary treatments or procedures. Equation (13) calculates the precision.

$$\text{Precision} = \frac{TP}{TP + FP} \tag{13}$$

- Recall, also known as sensitivity, is the ratio of $TP$ results divided by the number of all samples that should be identified as positive. It measures how well a model can identify all positive instances. Equation (14) calculates the recall.

$$\text{Recall} = \frac{TP}{TP + FN} \tag{14}$$

- F1-Score: F1-Score is the harmonic mean of precision and recall. It balances both metrics and is particularly useful when dealing with imbalanced datasets. Equation (15) calculates the F1-Score.

$$\text{F1-Score} = 2 * \frac{\text{Precision} * \text{Recall}}{T\text{Precision} + \text{Reca}} \tag{15}$$

- Loss function trains neural networks for accurate multi-class classification by minimizing the discrepancy between predicted probabilities and true labels. Equation (16) calculates the loss function.

$$\text{Loss function:} = -\sum_i y_{\text{true},i} \cdot \log\left(y_{\text{pred},i}\right) \tag{16}$$

*4.1. Experiment Result*

In this section, the results of the proposed model are analyzed using the olive disease dataset. The analysis of the results was used to check the validation of our model and to determine how it handles the processing and accurate prediction of the disease in the leaves of olive trees.

At the beginning of the discussion of the results, it is clear that the proposed system relies heavily on clustering. Figure 7 illustrates the number of clusters after each stage of the evolving process of the DECS algorithm.

The dataset includes three classes, meaning there should be three global clusters. However, applying the DECS algorithm results in seven clusters, which is acceptably close to the actual number of classes in the dataset. In its final evolving stage, the DECS algorithm achieved a silhouette score of 0.7. According to the number of clusters produced by DECS, the proposed system trained seven deep learning networks using those clusters.

Generally, the proposed model includes image enhancement, deep learning, and clustering techniques. Image enhancement and clustering techniques are integrated into the deep learning framework to constitute the system under consideration. Establishing a performance baseline is imperative to ascertain the significance of each component; this

baseline will subsequently function as a benchmark in the ablation study. Table 2 shows the ablation study of the components of the proposed model.

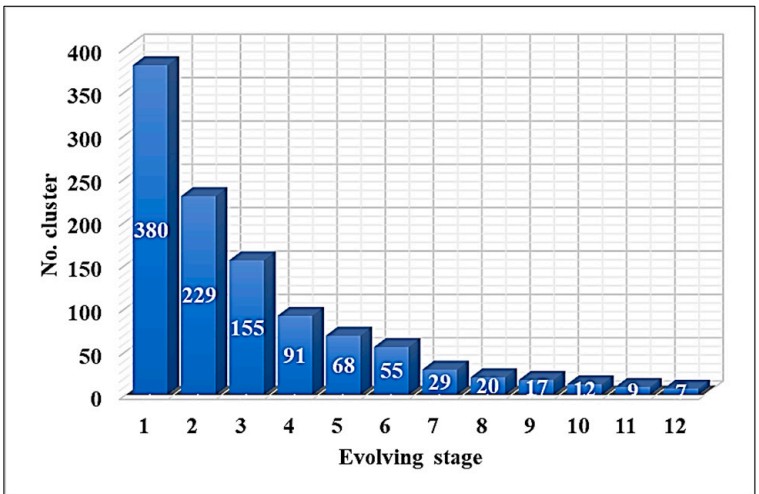

**Figure 7.** Clusters of DECS algorithms.

**Table 2.** Case study of the effects of the various combinations on the proposed model steps.

| Type | Component of the Proposed Model | | | Accuracy (%) |
|---|---|---|---|---|
| | Image Enhancement | Clustering | Deep Learning | |
| Combination I | No | No | Yes | 84.16 |
| Combination II | Yes | No | Yes | 94.09 |
| Combination III | No | Yes | Yes | 88.42 |
| Combination IV | Yes | Yes | Yes | 98.30 |

Table 2 presents the outcomes of a comprehensive ablation study conducted on the components of the proposed computational model. This investigative endeavor juxtaposes the efficacy of four distinct permutations, each incorporating varying combinations of image enhancement, clustering algorithms, and deep learning techniques, by assessing their corresponding accuracies. Combination I, exclusively reliant on deep learning methodologies, yielded an accuracy measure of 84.16%. This suggests that a standalone deep learning approach needs to achieve the requisite high accuracy. Conversely, Combination II, encompassing image enhancement and deep learning but eschewing clustering, secured a significantly higher accuracy of 94.09%. Notably, the image enhancement component refined edge delineation, thus improving the overall effectiveness of the deep learning algorithms compared to Combination I. In the case of Combination III, which incorporates clustering and deep learning while omitting image enhancement, an accuracy of 88.42% was attained. The clustering mechanism effectively reduces the cardinality of categories within each respective cluster, thereby augmenting the precision of deep learning-based predictions. However, the absence of image enhancement results in data characterized by noise and suboptimal feature delineation, thereby impeding categorical differentiation. Finally, our findings unequivocally demonstrate that Combination IV, which amalgamates all three components above, manifests the pinnacle of accuracy at 98.30%. These empirical results substantiate the assertion that integrating image enhancement, clustering, and deep learning into the proposed model culminates in optimal accuracy.

The proposed model is compared with the VGG16 and AlexNet deep learning algorithms concerning the loss function and accuracy. Figures 8–10 illustrate the comparative analysis among VGG16, AlexNet, and the proposed model. They show an examination of overfitting and evaluate convergence performance.

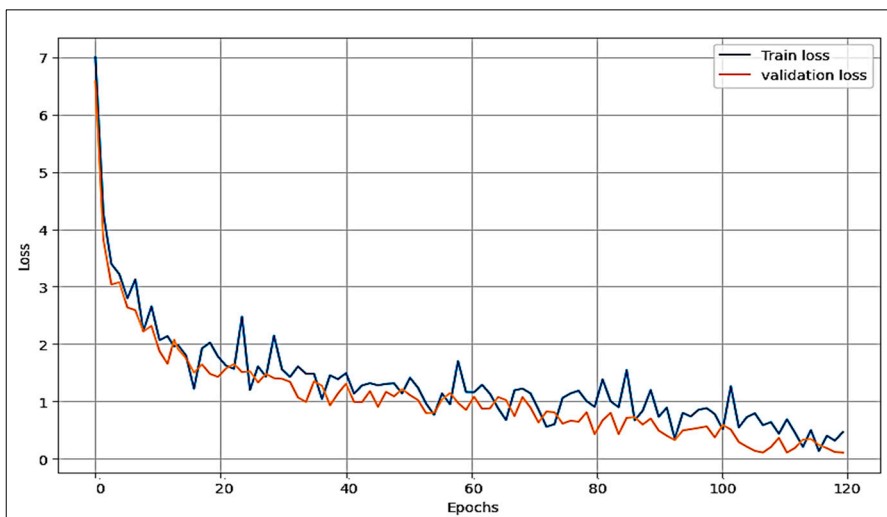

**Figure 8.** Performance of the proposed model (train and validation) on the loss function.

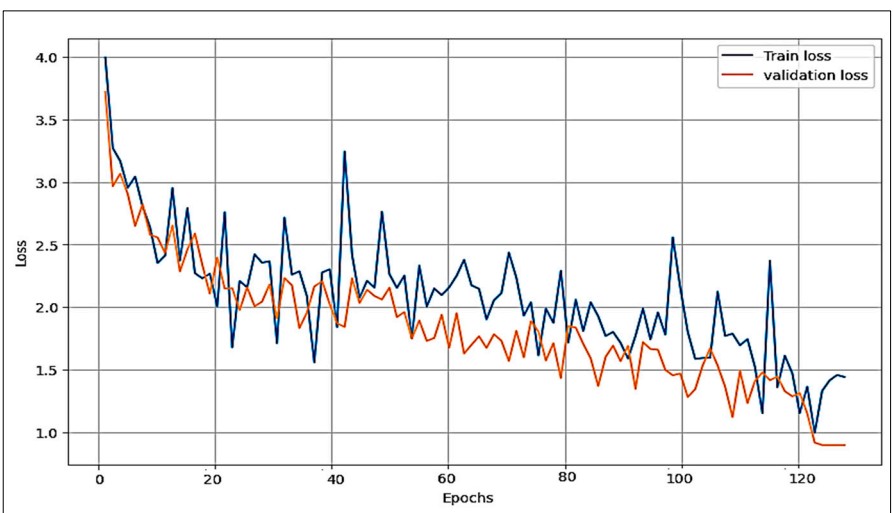

**Figure 9.** Performance of VGG16 (train and validation) on the loss function.

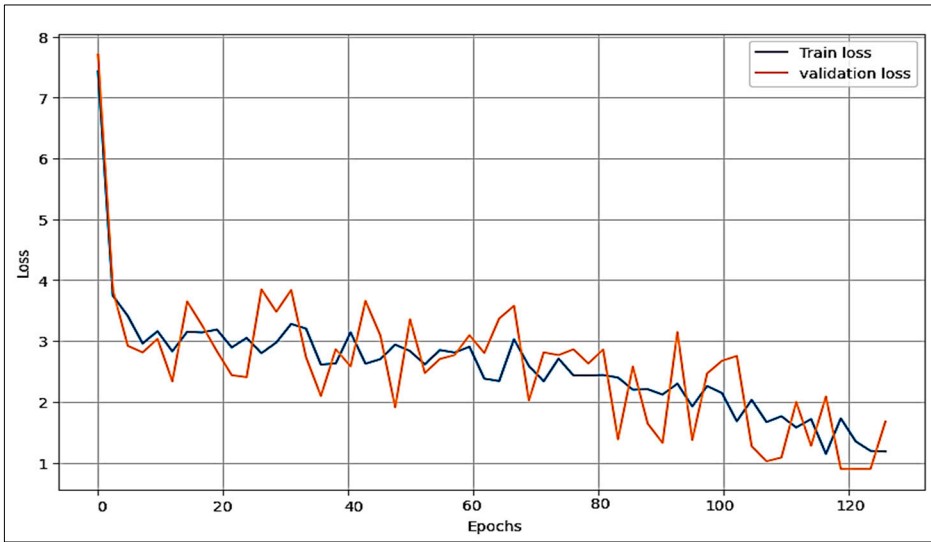

**Figure 10.** Performance of AlexNet (train and validation) on the loss function.

In the three figures presented, it is evident that both the proposed model and VGG16 show no signs of overfitting, demonstrating their robust generalization capabilities. However, a notable contrast arises in the case of AlexNet, where clear indications of overfitting are observed. This discrepancy highlights the ability of the proposed model and VGG16 to maintain a balance between training and validation performance. Comparing the proposed model and VGG16, it is clear that our model performs best. Of note is the remarkably smooth behaviour of the loss of function in our model, combined with a robust acquisition profile. These observations highlight our proposed approach's superior convergence ability and overall excellence over VGG16.

Table 3 compares the proposed model, AlexNet, and VGG16 performance metrics in the context of a classification task. The evaluation criteria include accuracy, precision, recall, and F1-score and provide a holistic perspective on the effectiveness of the models.

**Table 3.** Comparative performance metrics of proposed model, AlexNet, and VGG16.

| Model | Accuracy (%) | Precision | Recall | F1-Score |
| --- | --- | --- | --- | --- |
| Proposed | 98.3 | 98.5 | 98.2 | 98.3 |
| AlexNet | 95.0 | 95.4 | 95.8 | 95.6 |
| VGG16 | 96.0 | 96.1 | 95.7 | 95.9 |

It seems that the proposed model is performing very well, with an accuracy of 98.3% and high values for precision, recall, and F1 score. It is great to see that it can capture relevant instances while minimizing false positives and negatives. AlexNet and VGG16 also seem to perform well, although with slightly lower precision. It is interesting to see the comparative analysis and how the proposed model has a superior balance between accuracy and precision/recall.

Figure 11 displays three distinct confusion matrices, each corresponding to a different model: proposed model, AlexNet, and VGG16. These matrices illustrate the distribution of predicted classes against actual classes, highlighting true positives, true negatives, false positives, and false negatives for each model's predictions. The visual representation offers a clear overview of how effectively each model performs in classifying instances and identifying potential areas of improvement or optimization.

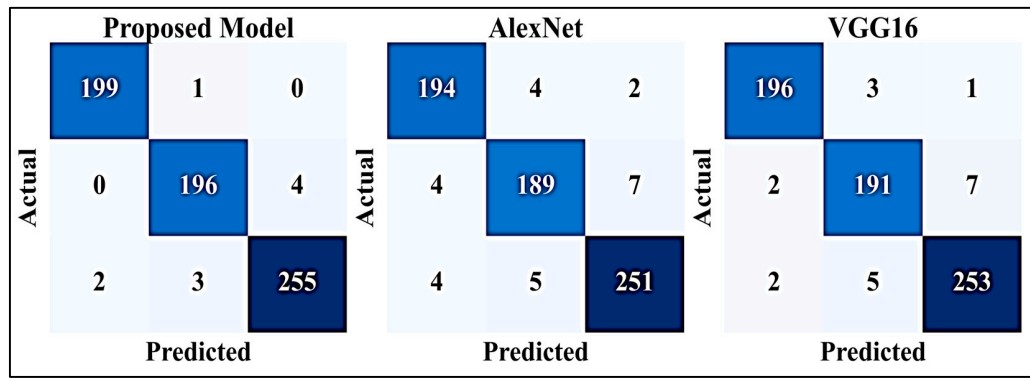

**Figure 11.** Confusion matrices for proposed model, AlexNet, and VGG16.

Figure 12 shows the comparative performance of the precision–recall curves for the three models: roposed model, AlexNet, and VGG16. These curves illustrate the interplay between precision and recall at different classification thresholds. The proposed model has a high precision and recall values curve, meaning it can identify positive instances while accurately minimizing false positives. The AlexNet and VGG16 curves show competing but slightly lower precision–recall values, highlighting their performance in capturing relevant samples. The visualization provides a nuanced perspective on the models' efficiency in handling precision–recall dynamics.

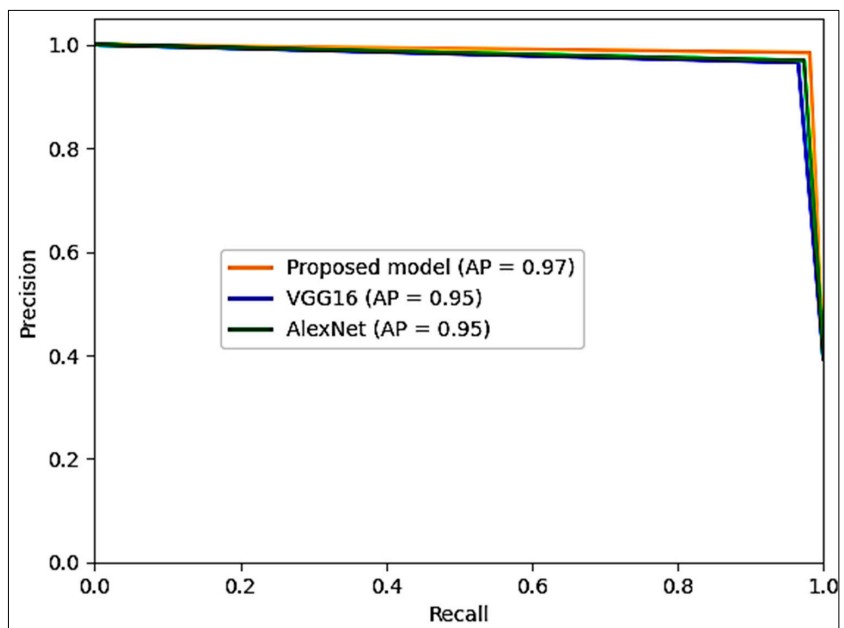

**Figure 12.** Confusion matrices for the proposed model, AlexNet, and VGG16.

Figure 13 compares receiver operating characteristic (ROC) curves for the proposed model, VGG16, and AlexNet. Each curve demonstrates the performance of the models in discriminating between positive and negative instances at different classification thresholds. The blue curve representing VGG16 and the green curve representing AlexNet show competitive performances with ROC. The expansive value ROC reflects the exceptional ability of the proposed model to discriminate between positive and negative instances at different thresholds. This remarkable discriminative ability positions it ahead of AlexNet and VGG16 and highlights its ability to classify positive instances while minimizing misclassifications correctly.

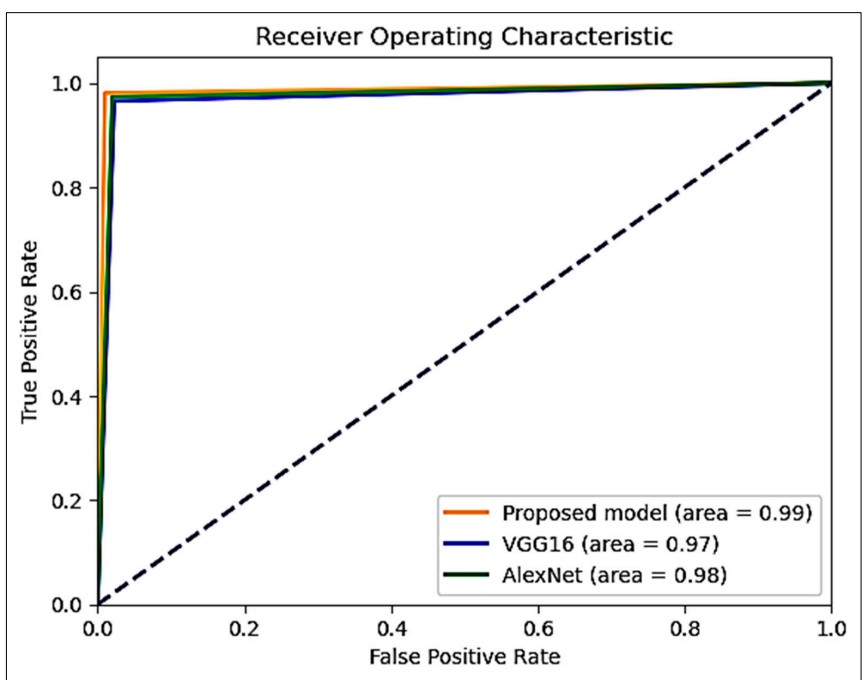

**Figure 13.** ROC metrics for the proposed model, AlexNet, and VGG16.

The clustering process significantly strengthens the system, resulting in improved performance. This approach harnesses the power of grouping similar elements and results

in a remarkable improvement in overall effectiveness. The incorporation of clustering techniques leads to a significant increase in performance capabilities.

### 4.2. Comperive with Other Studies

In a comparative analysis with other studies, the proposed model is outstanding, with an impressive accuracy of 98.3%. This remarkable performance places it in a favorable position alongside existing research efforts. In [4] and [13], they reported an accuracy of 96%, closely followed by [12], which achieved a commendable accuracy of 95.6%. While [10] attained an accuracy of 95%, the significant lead of the proposed model with an accuracy of 98.3% shows its potential for higher precision in classification tasks. In addition to its high accuracy, the proposed model achieved the best precision, recall, and F1 score, as shown in Table 4.

**Table 4.** Comparative performance with other studies.

| Ref. | Model Name | Populations year | Accuracy | Precision | Recall | F1-Score |
|---|---|---|---|---|---|---|
| [4] | • Olive disease classification based on vision transformer and CNN models | 2022 | 96% | 97% | 96% | 96% |
| [10] | • Classification of olive leaf diseases using deep convolutional neural networks | 2020 | 95% | 93% | 90% | 91% |
| [12] | • MobiRes-Net: A hybrid deep learning model for detecting and classifying olive leaf diseases | 2022 | 95.6 | 96.6% | 97.1% | 96.8% |
| [13] | • Olive leaf disease identification framework using Inception V3 deep learning | 2022 | 96% | Not mention | | |
| | • Proposed model | | 98.3% | 98.5 | 98.2 | 98.3 |

The proposed model's accuracy advantage is a testament to its robust design and sophisticated algorithms. This significant advantage positions it as a strong contender for applications that require precision-oriented results. The remarkable performance confirms the model's effectiveness and underscores its potential to exceed or at least match the accuracy levels achieved in other studies. Such performances underline the relevance and applicability of the proposed model in various domains and make it an enticing option for practitioners seeking optimal results in their classification efforts.

### 5. Conclusions

The cultivation of olive trees offers significant economic and health benefits. However, these trees are vulnerable to various diseases that can adversely affect the yield. When olive trees are afflicted with diseases, specific changes can be observed in the leaves, such as a fading of green color or the appearance of brown spots on the leaf surface. Based on color analysis, intelligent systems were developed to address this issue that classify olive leaf diseases. Therefore, image enhancement is necessary before entering deep learning or machine learning. However, this enhancement must be thoughtful when dealing with olive tree diseases to ensure no additional colors appear in the image. Training a deep learning network is crucial in improving performance and requires making many important decisions. The amount of data used in preparing the network must be appropriate for the architecture of the network. If the data are too much or too little, the network may suffer from overfitting, significantly impacting its performance. The number of classes in the data potentially affects the accuracy of the network. Therefore, this paper proposes a model for diagnosing diseases in olive tree leaves using a hybrid technique that combines deep learning and clustering. The initial phase encompasses color correction, guaranteeing fidelity in the representation image to the next steps. Following this, the images traverse through the prepared data to the extraction feature module, where they undergo normalization and partitioning, all in preparation for integration into a profound deep learning-based

classification network. The proposed model aims to enhance the performance of deep learning algorithms, achieving high accuracy while minimizing the tendency for overfitting. The clustering in the proposed model reduces overfitting and increases accuracy because it reduces the training data to fit the network structure and the number of categories in each cluster. The performance of the proposed model depends heavily on the data quality used for training. Practical usage of our proposed model can be seen in real-world applications where early detection and classification of diseases can significantly improve yield and reduce losses in olive cultivation. By providing accurate and timely diagnosis, our model can help farmers take necessary actions to prevent disease spread and ensure the healthy growth of their crops. For future works, we suggest exploring other deep learning architectures or techniques, such as LIME and SHAP, to improve performance and accuracy in detecting and classifying olive leaf diseases.

**Author Contributions:** A.H.A.: methodology, and writing review and editing. A.M.A.-j.: conceptualization. H.H.R.A.-M.: funding acquisition. S.M.H.: project administration. H.J.M.: investigation and visualization. M.R.A.: data curation and software. M.A.: formal analysis. R.R.N.: validation. All authors have read and agreed to the published version of the manuscript.

**Funding:** This research received no external funding.

**Institutional Review Board Statement:** Not applicable.

**Informed Consent Statement:** Not applicable.

**Data Availability Statement:** The data that support the findings of this study are openly available in the link: https://github.com/sinanuguz/CNN_olive_dataset (accessed 11 September 2023).

**Conflicts of Interest:** The authors declare no conflict of interest.

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
