# Peer review of "Dynamic Clustering Strategies Boosting Deep Learning in Olive Leaf Disease Diagnosis"

_sustainability, doi:10.3390/su151813723_

Round 1

Reviewer 1 Report

see the attachment 

Author Response

Dear Reviewer

Hello! I hope you are doing well. We are grateful for your thoughtful and thorough review of my manuscript titled “Dynamic Clustering Strategies Boosting Deep Learning in Olive Disease Diagnosis.” Your comments and suggestions have helped me improve the quality and impact of my research. We appreciate the time and effort you spent on this review.

We carefully considered your comments because they reflect your expertise and attention to detail. Your feedback aligns perfectly with my objective of contributing to artificial intelligence, data analysis, and machine learning. We are committed to addressing your points to improve the manuscript’s clarity, comprehensiveness, and scholarly merit.

We are uploading (a) our point-by-point response to the comments below, (b) the full updated manuscript with the changes highlighted (highlight in green regarding your comments, while the rest of the colors are according to other reviewers’ comments), and © a clean updated manuscript without highlights.

Note:

Reviewer#1 highlighted- Green

Reviewer#2 highlighted-Pink 

Reviewer#3 highlighted-Turquoise 

Reviewer#4 highlighted-yellow 

Note2:

We have made significant changes to the structure of the manuscript in response to the feedback provided by the reviewers. Our approach involved consolidating the comments made by all four reviewers and adopting a unified perspective that addresses all of their viewpoints.

Reviewer 2 Report

The authors proposed a Dynamic Clustering Strategies Boosting Deep Learning in Olive Disease Diagnosis which is found to be an interesting research area for the industry. However, the structure of this manuscript in particular needs extensive improvement in the following ways

1.     1.3. Evaluation strategies are not needed here. Delete it. Secondly, there is no need for 1.4. Paper organization to be a subheading. It should be a normal paragraph.

2.     A standard conventional journal paper format includes A. Introduction, B. Related studies, C. Materials and Methods, D. Result and Analysis, E. Conclusion. On that note the heading 3. Convolutional neural networks (CNN) and 4. Dynamic Clustering Empowering Deep Learning should be under 3. Materials and Method, with the subheading 3.1 Prerequisites. then 3.1.1 Convolutional neural networks (CNN), 3.1.2 Dynamic Clustering Empowering Deep Learning, etc.3.2 should now follow the proposed model, 3.3 Dataset and data preprocessing, 3.4 Evaluation metrics, etc.

3.     5.2 Prepared data should come after 5.3. Cluster train_data based on DECS, 5.5. Select a cluster

4.     Dataset and prepared dataset should be combined and be written as a subheading 3.3

5.     Check the subheading numbering again… before and after page 376

6.     Evaluation metrics should not be part of the result

7.     Comparative Performance with other studies should not be only accuracy… sensitivity, specificity, f1 score, and precision should be calculated and compared with

8.     The workload in this manuscript is very shallow. in other to make the manuscript more rich in content and for the good of the research community, explore all other deep learning model result explanations such as LIME AND SHAP

Minor revision

Author Response

Dear Reviewer

Hello! I hope you are doing well. We are grateful for your thoughtful and thorough review of my manuscript titled “Dynamic Clustering Strategies Boosting Deep Learning in Olive Disease Diagnosis.” Your comments and suggestions have helped me improve the quality and impact of my research. We appreciate the time and effort you spent on this review.

We carefully considered your comments because they reflect your expertise and attention to detail. Your feedback aligns perfectly with my objective of contributing to artificial intelligence, data analysis, and machine learning. We are committed to addressing your points to improve the manuscript’s clarity, comprehensiveness, and scholarly merit.

We are uploading (a) our point-by-point response to the comments below, (b) the full updated manuscript with the changes highlighted (highlight in Pink regarding your comments, while the rest of the colors are according to other reviewers’ comments), and © a clean updated manuscript without highlights.

Note:

Reviewer#1 highlighted- Green

Reviewer#2 highlighted-Pink 

Reviewer#3 highlighted-Turquoise 

Reviewer#4 highlighted-yellow 

Note2:

We have made significant changes to the structure of the manuscript in response to the feedback provided by the reviewers. Our approach involved consolidating the comments made by all four reviewers and adopting a unified perspective that addresses all of their viewpoints.

Reviewer 3 Report

The paper approaches a challenging topic referring to recent developments concerning an innovative model for olive disease diagnosis, combining AI tools for reliable results.

1.Introduction

We suggest the authors synthesize in one phrase the aim of their study.

Line 137 – we suggest the authors to replace healthcare with plants health care.

6. Results

6.1. Dataset

Line 433 – you mention leaves affected by the attack of Aculus olearius Castagnoli, 1977. We suggest the authors to include in Introduction section a discussion about the mite and motivate why do you choose this mite as exemplification of your disease strategy diagnosis.

Author Response

Dear Reviewer

Hello! I hope you are doing well. We are grateful for your thoughtful and thorough review of my manuscript titled “Dynamic Clustering Strategies Boosting Deep Learning in Olive Disease Diagnosis.” Your comments and suggestions have helped me improve the quality and impact of my research. We appreciate the time and effort you spent on this review.

We carefully considered your comments because they reflect your expertise and attention to detail. Your feedback aligns perfectly with my objective of contributing to artificial intelligence, data analysis, and machine learning. We are committed to addressing your points to improve the manuscript’s clarity, comprehensiveness, and scholarly merit.

We are uploading (a) our point-by-point response to the comments below, (b) the full updated manuscript with the changes highlighted (highlight in Turquoise regarding your comments, while the rest of the colors are according to other reviewers’ comments), and © a clean updated manuscript without highlights.

Note:

Reviewer#1 highlighted- Green

Reviewer#2 highlighted-Pink 

Reviewer#3 highlighted-Turquoise 

Reviewer#4 highlighted-yellow 

Note2:

We have made significant changes to the structure of the manuscript in response to the feedback provided by the reviewers. Our approach involved consolidating the comments made by all four reviewers and adopting a unified perspective that addresses all of their viewpoints.

Reviewer 4 Report

This manuscript presents a new diagnostic method for olive diseases. On the whole, the technique of this manuscript is novel, the data are sufficient, the results are credible, and it has certain significance for agricultural production. However, there are some problems with the manuscript

1. As can be seen from the content, the author studied the deep learning diagnosis method of olive leaf disease. I think the current title is not accurate enough, and the word leaf should be added.

2. Abstract introduces too much background knowledge, should add the introduction of methods, compared with what methods, and the overall accuracy situation.

3. The introduction needs to be rewritten. At present, many articles on the diagnosis of leaf pests and diseases have been published, so it is suggested that the author conduct a more reasonable review. In particular, why the use of deep clustering methods is proposed.

4. The author's table of contents needs to be reorganized, and there are too many duplicate section headings

5. I did not find the discussion of the article, so I suggest the author increase the discussion

This manuscript presents a new diagnostic method for olive diseases. On the whole, the technique of this manuscript is novel, the data are sufficient, the results are credible, and it has certain significance for agricultural production. However, there are some problems with the manuscript

1. As can be seen from the content, the author studied the deep learning diagnosis method of olive leaf disease. I think the current title is not accurate enough, and the word leaf should be added.

2. Abstract introduces too much background knowledge, should add the introduction of methods, compared with what methods, and the overall accuracy situation.

3. The introduction needs to be rewritten. At present, many articles on the diagnosis of leaf pests and diseases have been published, so it is suggested that the author conduct a more reasonable review. In particular, why the use of deep clustering methods is proposed.

4. The author's table of contents needs to be reorganized, and there are too many duplicate section headings

5. I did not find the discussion of the article, so I suggest the author increase the discussion

Author Response

Dear Reviewer

Hello! I hope you are doing well. We are grateful for your thoughtful and thorough review of my manuscript titled “Dynamic Clustering Strategies Boosting Deep Learning in Olive Disease Diagnosis.” Your comments and suggestions have helped me improve the quality and impact of my research. We appreciate the time and effort you spent on this review.

We carefully considered your comments because they reflect your expertise and attention to detail. Your feedback aligns perfectly with my objective of contributing to artificial intelligence, data analysis, and machine learning. We are committed to addressing your points to improve the manuscript’s clarity, comprehensiveness, and scholarly merit.

We are uploading (a) our point-by-point response to the comments below, (b) the full updated manuscript with the changes highlighted (highlight in yellow regarding your comments, while the rest of the colors are according to other reviewers’ comments), and © a clean updated manuscript without highlights.

Note:

Reviewer#1 highlighted- Green

Reviewer#2 highlighted-Pink 

Reviewer#3 highlighted-Turquoise 

Reviewer#4 highlighted-yellow 

Note2:

We have made significant changes to the structure of the manuscript in response to the feedback provided by the reviewers. Our approach involved consolidating the comments made by all four reviewers and adopting a unified perspective that addresses all of their viewpoints.

Round 2

Reviewer 1 Report

i have seen that the author has incorporated most of the comments but there are some comments which are were not properly addressed such as 

1. Practical usage of the proposed work is not mentioned neither in the introduction nor in the conclusion. 

2. But fails to mention how the proposed model has achieved such accuracy. It is a must to perform an ablation study.?

3. number of figures and tables is wrong throughout the manuscript. the author must double-check the numbering as well as cross-reference within the text.

Minor editing of English language required

Author Response

Dear Reviewer,

I hope this message finds you well. Thank you very much for reviewing our manuscript,”  Dynamic Clustering Strategies Boosting Deep Learning in   Olive Leaf Disease Diagnosis". We appreciate your thoughtful comments, which have guided us in refining and improving our work.

We have prepared a point-by-point response to your comments, explaining the changes made or reasons for retaining certain sections.

An updated version of the manuscript has been provided, with all the changes made in response to your comments highlighted in green for easy reference. Modifications made due to other reviewers' feedback have been highlighted using different colors.

A clean, updated manuscript version has also been uploaded for your convenience without any highlights.

Specifically, we have focused on addressing the three main points you raised:

  1. Practical Usage of Proposed Work: We have included in the conclusion that explicitly outlines our work's practical applications and benefits. This provides context and underscores the value of our research.
  2. Ablation Study for Model Accuracy: We have conducted an ablation study to clarify how our model achieved its reported accuracy. The results are incorporated into the manuscript, offering insights into the contributions of different components of our model.
  3. Numbering of Figures and Tables: We have carefully revised the manuscript to correct the numbering of figures and tables and ensured their proper cross-referencing within the text.
  4. As you may note, we have made further edits to the manuscript to improve the English language. These changes are also highlighted in turquoise for your easy reference.

We hope that these revisions adequately address your comments and concerns. Your feedback has been invaluable in enhancing the quality of our manuscript, and we are optimistic that these changes will meet your approval.

Best Regards

Ali H. Alsaeedi (on behalf of the authors)

Reviewer 2 Report

Not Applicable

Author Response

Dear Reviewer,

We hope this message finds you well. First and foremost, I would like to express my sincere gratitude for your in-depth and constructive review of my manuscript, "Dynamic Clustering Strategies Boosting Deep Learning in Olive Disease Diagnosis." Your valuable comments and suggestions have been instrumental in enhancing the quality and impact of my work. Your time and effort spent on this review are highly appreciated.

We'd also like to apologize if our responses during the first review cycle were not as clear as they could have been. The feedback from multiple reviewers occasionally led to overlapping modifications, which might have made our revisions seem ambiguous or inconsistent. We have made extra efforts this time to clarify any such issues.

In response to your feedback, we have taken the following steps:

  1. We have prepared a point-by-point response to your comments, explaining the changes made or reasons for retaining certain sections.
  2. An updated version of the manuscript has been provided, with all the changes made in response to your comments highlighted in yellow for easy reference. Modifications made due to other reviewers' feedback have been highlighted using different colors.
  3. A clean, updated version of the manuscript has also been uploaded for your convenience without any highlights.

We sincerely hope the revisions are to your satisfaction and look forward to your thoughts on the updated manuscript.

Best Regards

Ali H. Alsaeedi (on behalf of the authors)

Reviewer 4 Report

I have no more questions.

Author Response

Dear Reviewer,

I hope this message finds you well. I am pleased to hear that all the revisions made according to your instructions have been satisfactory. This greatly encourages me and boosts my confidence in the research conducted.

I look forward to receiving more of your constructive comments in the future. Thank you once again for your support and guidance.

Best Regards 

Ali H. Alsaeedi (on behalf of the authors)